# Cost-benefit tradeoff mediates the transition from rule-based to memory-based processing during practice

**Guochun Yang** [1,2]*, **Jiefeng Jiang** [1,2,3]*

**1** Cognitive Control Collaborative, University of Iowa, Iowa City, Iowa, United States of America,
**2** Department of Psychological and Brain Sciences, University of Iowa, Iowa City, Iowa, United States of America, **3** Iowa Neuroscience Institute, University of Iowa, Iowa City, Iowa, United States of America

\* guochun-yang@uiowa.edu (GY); jiefeng-jiang@uiowa.edu (JJ)

**Data Availability Statement:** All data needed to reproduce the Figs 2A, 4G, 5D, 6B, 7B, S3B and S6G in the paper are present in S1 Data. The raw neuroimaging data are publicly accessible at https://openneuro.org/datasets/ds005733/ (DOI:

## Abstract

Practice not only improves task performance but also changes task execution from rule- to memory-based processing by incorporating experiences from practice. However, *how* and *when* this change occurs is unclear. We test the hypothesis that strategy transitions in task learning can result from decision-making guided by cost-benefit analysis. Participants learn 2 task sequences and are then queried about the task type at a cued sequence and position. Behavioral improvement with practice can be accounted for by a computational model implementing cost-benefit analysis and the model-predicted strategy transition points align with the observed behavioral slowing. Model comparisons using behavioral data show that strategy transitions are better explained by a cost-benefit analysis across alternative strategies rather than solely on memory strength. Model-guided fMRI findings suggest that the brain encodes a decision variable reflecting the cost-benefit analysis and that different strategy representations are double-dissociated. Further analyses reveal that strategy transitions are associated with activation patterns in the dorsolateral prefrontal cortex and increased pattern separation in the ventromedial prefrontal cortex. Together, these findings support cost-benefit analysis as a mechanism of practice-induced strategy shift.

## Introduction

During task learning, practice improves performance and reduces effort. Practice effects occur in multiple cognitive functions supporting task execution. For example, it speeds up the processing of task-relevant information [1,2]. Additionally, with practice, task execution becomes more robust to distraction and requires less attentional control [3,4]. Crucially, practice also changes the strategy used to perform a task. For example, Schuck and colleagues [5] observed that with practice participants spontaneously shift from a rule-based strategy to a less effortful strategy relying on task-irrelevant information. More generally, strategy switches can occur between 2 common types of strategies: those based on task rules and those based on experiences [6–9]. For example, novice chess players must apply the rules to search for the best move, whereas experienced players can leverage their memory of previous games to recognize

10.18112/openneuro.ds005733.v1.0.1). The behavioral modeling code and data are available at https://zenodo.org/records/14397947 (DOI: 10.5281/zenodo.14397947).

**Funding:** This project was supported by the National Institute of Mental Health (https://www.nimh.nih.gov) (R01MH131559 to J.J.). This work was conducted on an MRI instrument funded by the National Institutes of Health (https://www.nih.gov) under grant number S10RR028821. The funders have played no role in the research.

**Competing interests:** The authors have declared that no competing interests exist.

**Abbreviations:** AIC, Akaike information criteria; CI, confidence interval; dlPFC, dorsolateral prefrontal cortex; EPI, echo-planar imaging; ER, error rate; FDR, false discovery rate; FWHM, full width at half maximum; GLM, generalized linear model; HRF, hemodynamic response function; LME, linear mixed-effect; MPFC, medial prefrontal cortex; RSS, residual sum of squares; RT, response time; SCEF, supplementary and cingulate eye field; vmPFC, ventromedial prefrontal cortex.

board configurations, thus reducing the effort of searching and retrieving the best move [10–12]. This transition from rule-based processing to memory-based retrieval is termed automatization and is vital in learning complex tasks (e.g., chess, programming, and arithmetic) [13,14]. However, little is known about *when* and *how* this transition happens.

Logan [13] proposes that the transition from rule- to memory-based processing can be conceptualized as a competition between the 2 strategies, with the winning strategy determined by which strategy completes the task more efficiently. Early in training, the rule-based strategy tends to dominate. With increased practice, the memory strategy becomes more advantageous and is more likely to prevail. This suggests a transition point after which performance would primarily rely on memory retrieval. A key question concerns the mechanisms underlying this transition. Logan posits that the transition could occur "*either as a consequence of statistical properties of the race or because of a strategic decision to trust memory and abandon the algorithm.*" [13] While Logan and colleagues [13,15] have extensively examined the former possibility, it remains unclear whether the transition can be explained through a decision-making framework. A previous study [5] found that signals in the medial prefrontal cortex (MPFC) predict the strategy transition ahead of time, suggesting the possibility of strategy evaluation and decision-making processes underlying the shift. However, systematic tests of this hypothesis are sparse.

The decision-making literature proposes that the control of goal-directed behavior relies on a cost-benefit analysis between alternative options [16,17]. Similarly, we hypothesize that the transitions in strategy during task performance may also reflect the optimization of a cost-benefit tradeoff. Conceptually, a memory-based strategy retrieves the answer directly and bypasses the execution of the rules. Thus, it is less effortful and less costly than a rule-based strategy. However, the memory-based strategy depends on the accumulation of experience, meaning its benefit is initially low and gradually increases as the memory becomes stronger with practice. In contrast, the benefit of rule-based processing remains consistently high from the outset. Therefore, early in task learning (e.g., a novice chess player), the rule-based strategy is favored because its higher benefit outweighs its higher cost compared to the memory-based strategy. As more experience is gained (e.g., a veteran chess player), the increased benefit enables the memory-based strategy to outperform the rule-based strategy in the cost-benefit tradeoff, leading to a strategy transition from rule- to memory-based processing. The memory-based strategy remains the winning strategy afterwards.

In this study, we aim to test the hypothesis that the strategy transition is determined by comparing values across alternative strategies. We call the model meeting these criteria the decision-selection model (Fig 1A, left). Alternatively, different strategies could be implemented simultaneously without the decision-making process, which we refer to as the both-active model (Fig 1A, middle). Another possible hypothesis is that the decision to use memory-based strategy could be determined solely by the strength of memories, with the transition occurring whenever the memories are strong enough to retrieve instances [18], independent of the value associated with the rule-based strategy. We term this approach the decision-memory-threshold model (Fig 1A, right). It is important to note that the decision-selection and both-active models share identical model settings because they predict the same behavioral patterns but differ in their predictions regarding neural representation (see below).

To test the decision-selection model, we design a two-phase experiment. Human participants (*n* = 33) learn 2 different task sequences (i.e., A and B) involving the same component tasks (i.e., Helmet, Vest, Glasses, Tool, and Weapon) in the training phase (Fig 1B; each task sequence consists of 5 distinct delayed matching tasks; see Materials and methods). In the following test phase, participants are asked to report the task type at a cued position from one sequence (e.g., A4, denoting the fourth position in sequence A) while undergoing fMRI

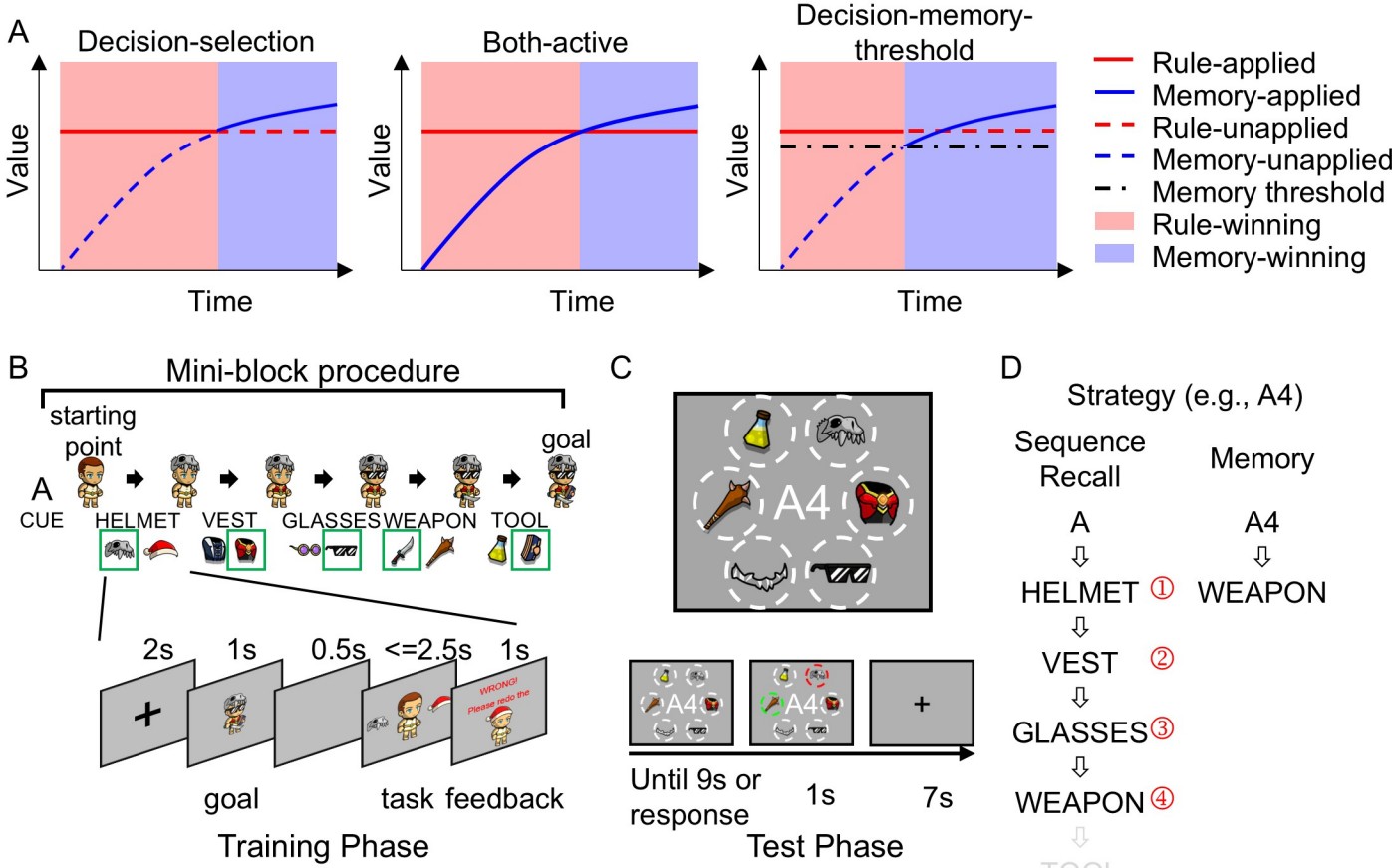

**Fig 1. Alternative models and the experimental design.** (A) Three major alternative models and how improved memory from practice affects strategy application. In the decision-selection model, the memory-based strategy replaces the rule-based strategy when the former provides better value (i.e., cost-benefit tradeoff). In the both-active model, both strategies are always implemented. In the decision-memory-threshold model, the strategy switches to memory-based whenever the memory strength reaches a predefined threshold. (B) Design of the training phase. Top panel: During the training phase, participants learn 2 sequences of tasks (one example sequence shown). Each sequence consists of 5 different tasks in a fixed order, indicated by a cue (A or B) presented before the first trial of the corresponding sequence. Each task requires the participant to equip an avatar with a specific type of gear (e.g., helmet). Bottom panel: A task starts with a goal image and then 2 option gears, one on each side of an avatar. Participants are required to choose the gear that matches the goal image. The goal image is randomized for each mini-block to ensure that the sequences are defined by equipment types (e.g., weapon) rather than by specific stimuli (e.g., dagger). (C) Design of the test phase. Top panel: During the test phase, a cue denoting the position from a sequence (e.g., A4, denoting the fourth position in sequence A) is displayed in the center of the screen. Surrounding the cue are 6 equipment images denoting 5 task types plus a foil (MOUSTACHE), with their spatial locations fixed but the display images randomly selected within each equipment type. Participants are required to identify the task type at the cued position of the sequence by selecting the equipment representing the task type (e.g., nunchaku, representing the weapon task). Bottom panel: trial time course. (D) Illustration of the 2 strategies. The rule-based strategy recalls the cued sequence (e.g., A) from the first position till the cued position (e.g., 4), whereas the memory strategy retrieves the association between the cue (e.g., A4) and the task (e.g., weapon) directly.

scanning (Fig 1C). As previously described, the decision-selection model predicts that participants will start with a rule-based strategy (i.e., by recalling the task sequence learned from the training phase and identifying the cued task) and switch to the memory-based strategy when the retrieval of cue-task associations becomes more efficient than sequence recall with practice (Fig 1D).

To preview the results, we construct a computational model (M0) to account for performance by determining the optimal strategy in a way consistent with both the decision-selection and the both-active models. We find that M0 can successfully fit the behavioral data and identify transition points marked by a switch cost occurring after the strategy transition. We also test several alternative models, including nested models (M1-M5) and extended models (M6-M8), and find that M0 outperforms all alternative models. Importantly, M0 also

outperforms the decision-memory-threshold model (M9) in accounting for behavioral data. To compare the decision-selection model with the both-active model, we focus on their different predictions on neural representations. Specifically, the decision-selection model posits that the brain encodes a decision variable reflecting the value difference between the rule- and memory-based strategies, whereas the both-active model does not assume the existence of such a decision variable. Our fMRI findings support the decision-selection model over the both-active model.

To gain insights into the neural substrates of the decision-selection model, we compare the fMRI patterns from trials where the model predicts different strategies and find that neural patterns show a double dissociation across a broad range of brain regions. We also analyze fMRI signal changes surrounding strategy transitions and find that the right dorsolateral prefrontal cortex (dlPFC) encodes the current winning strategy. Linking the improved memory to strategy transitions, we observe that the ventromedial PFC (vmPFC) tracks the increase of the memory strength of cue-task associations. In summary, these findings are consistent with the idea that strategy selection is driven by a dynamic cost-benefit analysis and that the strategy transition occurs when the value (i.e., cost-benefit tradeoff) of the memory-based strategy outweighs that of the rule-based strategy.

## Results

The results are organized into 2 parts: The first part compares the decision-selection model with alternative models (Fig 2 and Table 1). The second part investigates the neural substrates and their behavioral significance of the key components of the decision-selection model (Figs 3–7).

### Model-free behavioral results

The task involves training and test phases. In the training phase, participants were trained to learn the transition orders of 2 sequences (A and B) until they remembered transitions between tasks with greater than 90% accuracy (see Materials and methods). In the test phase, they were asked to select the task type of a sequence at a specific position (e.g., A4, indicating the fourth task of sequence A) within the scanner. For the test phase data, we conducted a model-free analysis. Since the rule-based strategy assumes a higher cost for later positions, whereas the memory strategy treats different cued positions equally, a transition from rule-based to memory-based strategy would predict a decreasing cost for later positions over time. To test this hypothesis, we analyzed trial-wise response times (RTs) and error rates (ERs) separately using linear mixed-effects (LME) models. Predictors included cued position (1–4), block (1–6), and their interaction. RT results (Fig 2A) showed a significant main effect of block ($\beta$ = −0.33, SE = 0.03, 95% confidence interval (CI) = [−0.39, −0.28], $t(33.3)$ = −11.41, $p < 0.001$, $\eta_p^2$ = 0.40), a significant main effect of position ($\beta$ = 0.48, SE = 0.05, 95% CI = [0.38, 0.59], $t(33.0)$ = 9.68, $p < 0.001$, $\eta_p^2$ = 0.58), and a significant interaction effect ($\beta$ = −0.10, SE = 0.02, 95% CI = [−0.14, −0.07], $t(59.8)$ = −5.73, $p < 0.001$, $\eta_p^2$ = 0.06). Post hoc comparisons showed that the linear effect of position in block 1 and 2 was larger than block 2–6 and 4–6, respectively ($p$s < 0.05). ER results showed significant main effects of block ($\beta$ = −0.20, SE = 0.05, 95% CI = [−0.29, −0.10], $t(33.0)$ = −4.17, $p < 0.001$, $\eta_p^2$ = 0.06), position ($\beta$ = 0.20, SE = 0.05, 95% CI = [0.11, 0.30], $t(36.6)$ = 4.41, $p < 0.001$, $\eta_p^2$ = 0.06), and a significant interaction effect ($\beta$ = −0.13, SE = 0.04, 95% CI = [−0.21, −0.04], $t(40.4)$ = −3.09, $p = 0.004$, $\eta_p^2$ = 0.02). Post hoc comparisons showed that the linear effect of position in block 1 was larger than in blocks 3 and 6, in block 2 compared to blocks 3, 4, and 6, and in block 5 compared to block 6, $p$s < 0.05. We also replicated these findings using all 5 cued positions (Note A in S1 Text). Note that the slope of

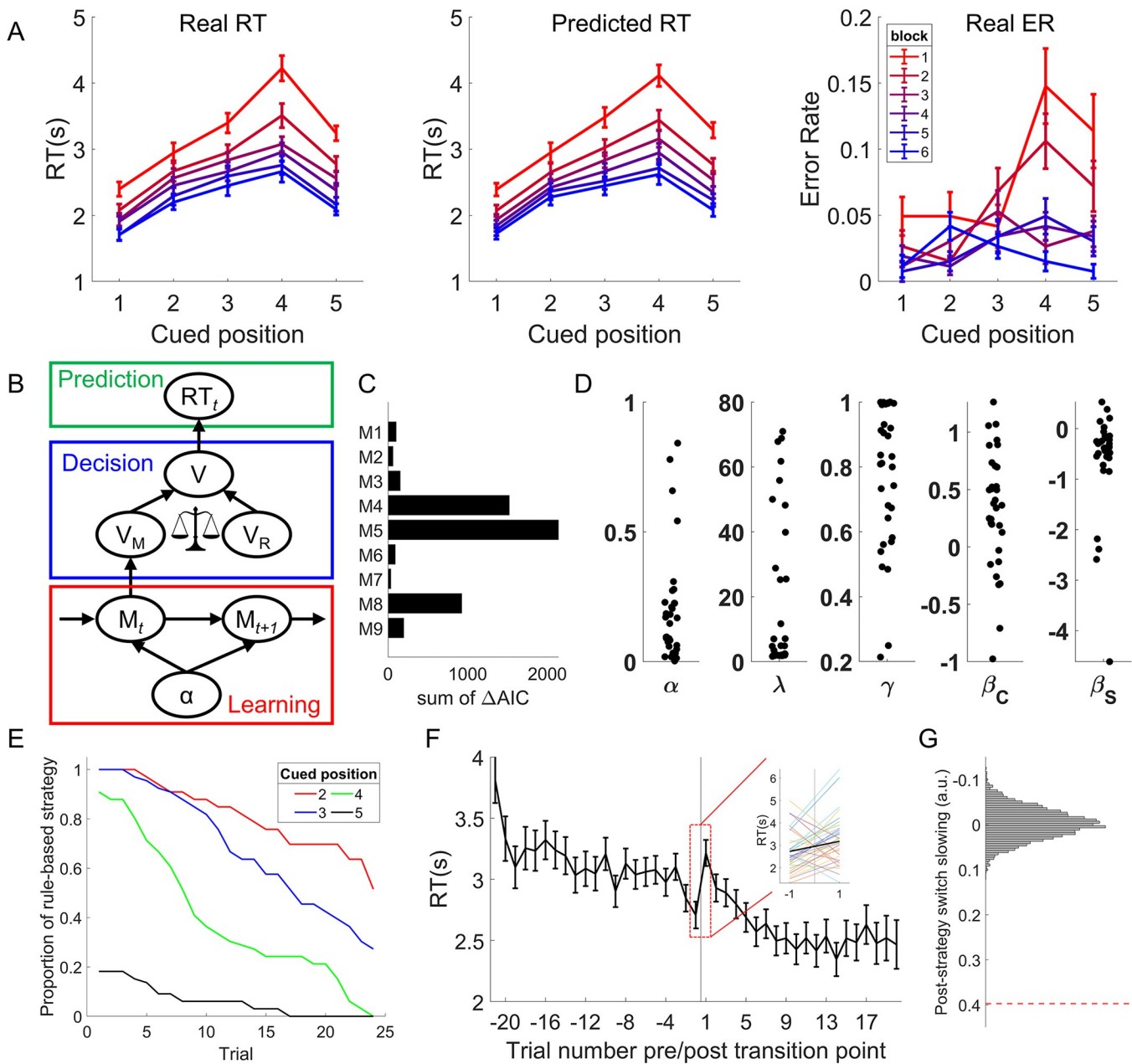

**Fig 2. Behavioral and computational modeling results.** (A) Behavioral data (RT, left panel, and ER, right panel) and model predicted RT (middle panel) displayed as a function of cued position and block. Error bars denote standard errors of the mean (SEMs). Data used for (A) can be found in S1 Data, specifically in the sheet labeled "Fig 2A." (B) The structure of the decision-selection model (M0). (C) Model comparison based on the group-level ΔAIC, i.e., increased AIC of alternative models (see Table 1 and Materials and methods) compared to the winning model (M0, illustrated in panel B). (D) Individual model-estimated parameters (α = learning rate, λ = cost-benefit tradeoff, γ = recency effect, see definitions in Materials and methods) and beta coefficients for rule implementation cost (C) and memory strength (S) effects. (E) Change of model predicted strategy over time. The Y axis shows the model-inferred proportion of participants using the rule-based strategy. Trials were counted for each cue separately. (F) The RT as a function of trial number relative to the cue's transition point. The inset highlights the increase of RT on the trial following the estimated strategy switch. Error bars denote SEMs. (G) Post-strategy switch slowing of real data (dashed red line) and its null distribution (histogram) estimated using randomly selected transition points. AIC, Akaike information criteria; ER, error rate; RT, response time.

block 6 was not completely flat because not all participants had transitioned to memory strategy by that time. Overall, we observed 3 patterns: (1) performance, when collapsed across cued positions, improved over time; (2) performance, when collapsed across blocks, was worse

**Table 1. Description of tested models and model comparison results.**

| Model regressors | M0* | M1 | M2 | M3 | M4 | M5 | M6 | M7 | M8 | M9 |
|---|---|---|---|---|---|---|---|---|---|---|
| C | + | + | + | + | + | − | − | + | + | + |
| C discounting | − | − | − | − | − | − | + | − | − | − |
| backward recall | − | − | − | − | − | − | − | + | − | − |
| S | + | + | + | + | − | − | + | + | + | + |
| Type (R/M) | + | − | + | − | − | − | + | + | + | + |
| C*Times$_R$ | + | + | − | − | + | − | + | + | + | + |
| Comparison between strategies | + | + | + | + | − | − | + | + | + | − |
| **P**robabilistic / **D**eterministic | D | D | D | D | D | D | D | D | P | D |
| AIC | −4,019 | −3,917 | −3,959 | −3,866 | −2,497 | −1,878 | −3,928 | −3,984 | −3,095 | −3,823 |
| Compared with M0 (*p*-value) | | <0.001 | <0.001 | <0.001 | <0.001 | <0.001 | <0.001 | <0.001 | <0.001 | <0.001 |

Note. The predictors of logged trial number and response were included in all models and are hence not reported in this table. C = cost of rule implementation (forward recall); S = memory strength; R = rule trial; M = memory trial; AIC = Akaike information criteria. Specifically, M0 represents settings of both the decision-selection and the both-active models, and M9 represents the decision-memory-threshold model.

* Denotes the winning model.

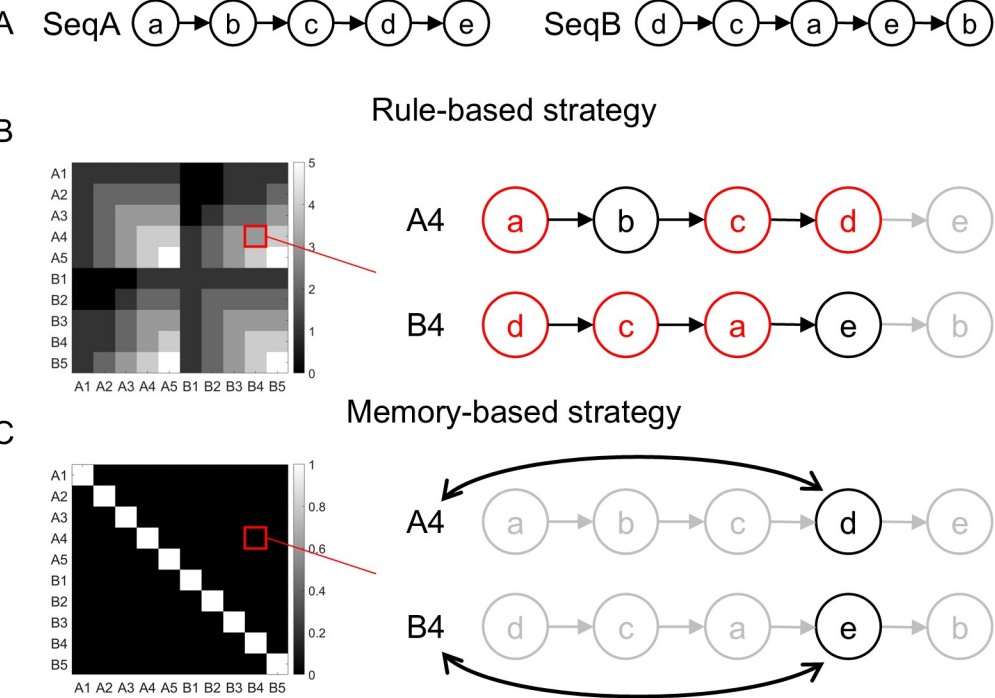

**Fig 3. Coding schemes of representational similarity for rule- and memory-based effects.** (A) The 2 task sequences. Letters a–e represent the 5 tasks. (B) The hypothetical representational similarity for the rule effect is quantified as the number of shared recalled tasks between the 2 trials. For example, cues A4 and B4 share 3 tasks in rule implementation, namely a, c, and d (red nodes), resulting in a similarity level of 3. (C) The hypothetical representational similarity for the cue effect (reflecting the memory strategy) is defined by the concordance of the cues and hence their associated tasks. For example, the cues A4 and B4 are different, thereby yielding a similarity value of 0. The light gray nodes and arrows in panels B and C indicate the tasks that are neither sequentially recalled nor retrieved by the cue.

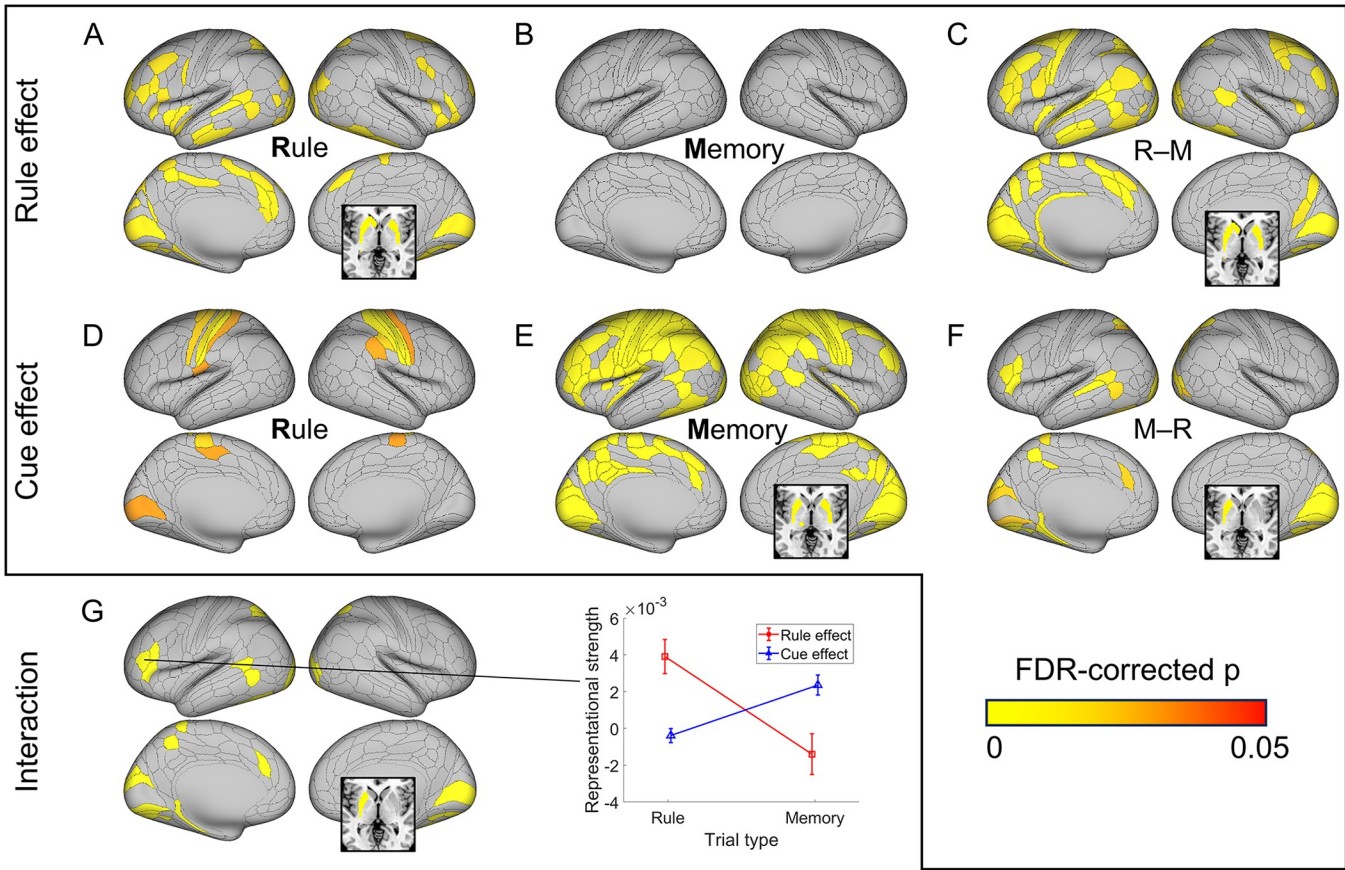

**Fig 4. Double dissociation of rule and memory representations on their respective trials revealed by multivariate activation patterns.** A significant representation of the rule effect is observed on rule (R) trials (panel A) but not on memory (M) trials (panel B). (C) A stronger rule effect on rule trials than memory trials (R–M) is found in frontoparietal, temporal, occipital, and subcortical regions. In contrast, the cue effect is observed in more regions on memory trials (panel E) than on rule trials (panel D). Regions showing a stronger memory effect on memory than on rule trials (M–R) include frontoparietal, temporal, occipital, and subcortical regions (panel F). (G) Regions showing double dissociation (i.e., the conjunction of C and F) with both stronger rule effect representations in rule trials and stronger cue effect representations in memory trials. The middle panel illustrates the double dissociation with an example region (left inferior prefrontal sulcus, anterior; IFSa). Error bars are standard errors of mean. Data used for the middle panel of (G) can be found in S1 Data, specifically in the sheet labeled "Fig 4G." All subcortical regions are depicted in an axial slice at z = 0.

when the later sequence positions were cued; and (3) this negative association between performance and cued position decreased with time.

## Model-based behavioral results

We constructed several computational models to capture the dynamics of transitioning from rule- to memory-based strategies (Fig 2B and Materials and methods). Our first model (M0) is a quantitative implementation of the decision-selection model. It includes 3 main components: learning, decision, and prediction. The learning component updates the memory strength of the cue-task associations (i.e., a probabilistic distribution of the correct task, denoted as M) using a delta rule with learning rate $\alpha$. The decision component makes winner-take-all decisions about which strategy to apply using cost-benefit analyses. The prediction component predicts trial-wise RT based on linear predictors of values driven by rule- ($V_R$) or memory-based ($V_M$) strategies, depending on the strategy used. The model was fit to trial-wise RT data to estimate model parameters and trial-wise predictions of the strategy used. On each trial, the model performs a cost-benefit analysis to determine which strategy to use based on their

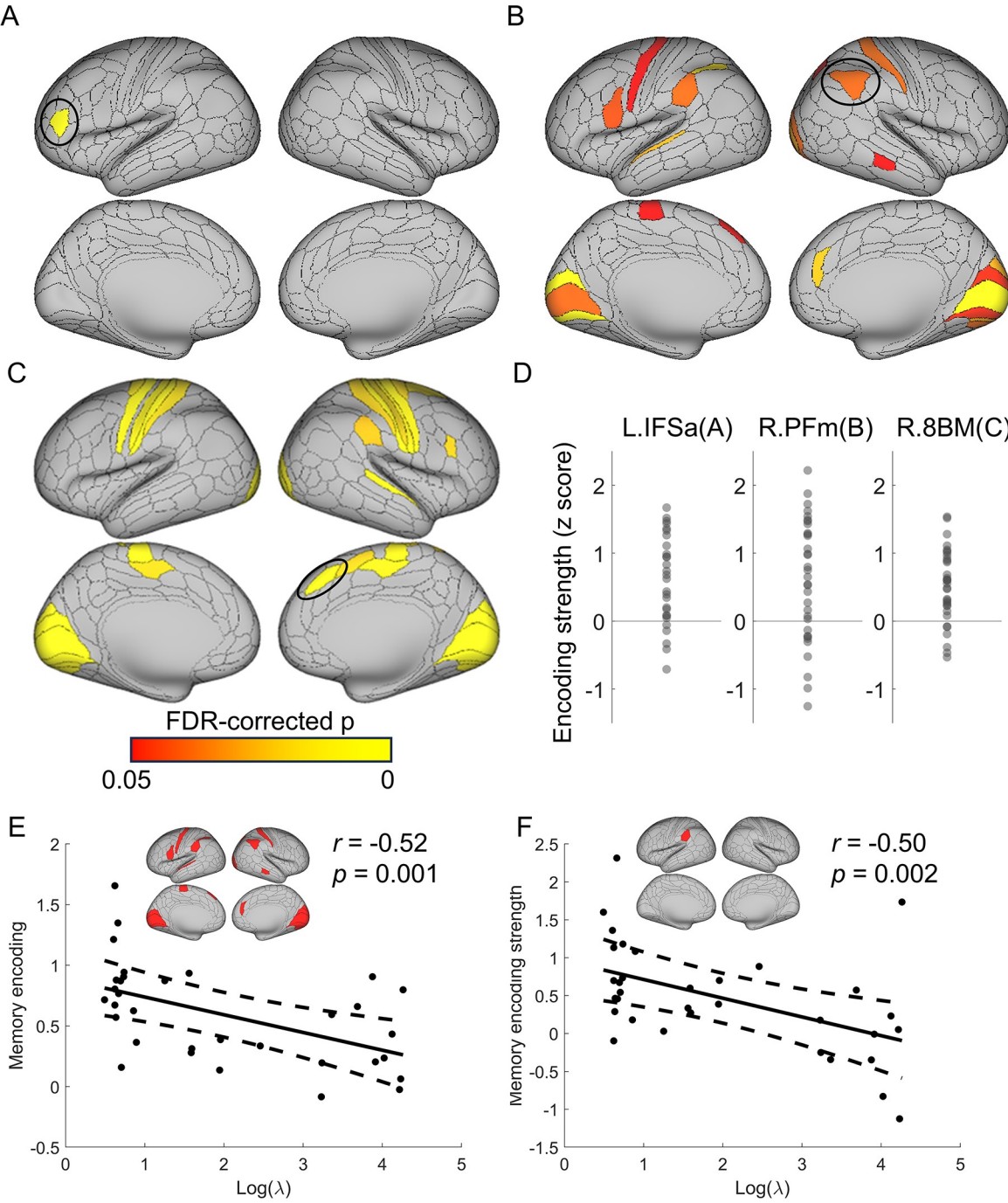

**Fig 5. Multivariate encoding of rule-, memory-based strategies and the decision variable.** (A) The left inferior prefrontal region encodes the cued position, reflecting the cost of rule implementation. (B) The memory strength is encoded in the visual, sensorimotor, and temporoparietal association regions. (C) The decision variable is encoded in sensorimotor and dorsomedial prefrontal regions. (D) The encoding strength of 3 representative regions (highlighted by black ovals in A–C) for the cued position (left), memory strength (middle), and decision variable (right). Each dot represents one subject. Data used for (D) can be found in S1 Data, specifically in the sheet labeled "Fig 5D." (E, F) Show a negative correlation between the model prediction of the cost-benefit tradeoff factor (log-transformed λ) and the neural memory encoding strength from those brain regions identified in (B) on average (E) and a single region left PF for example (F), respectively.

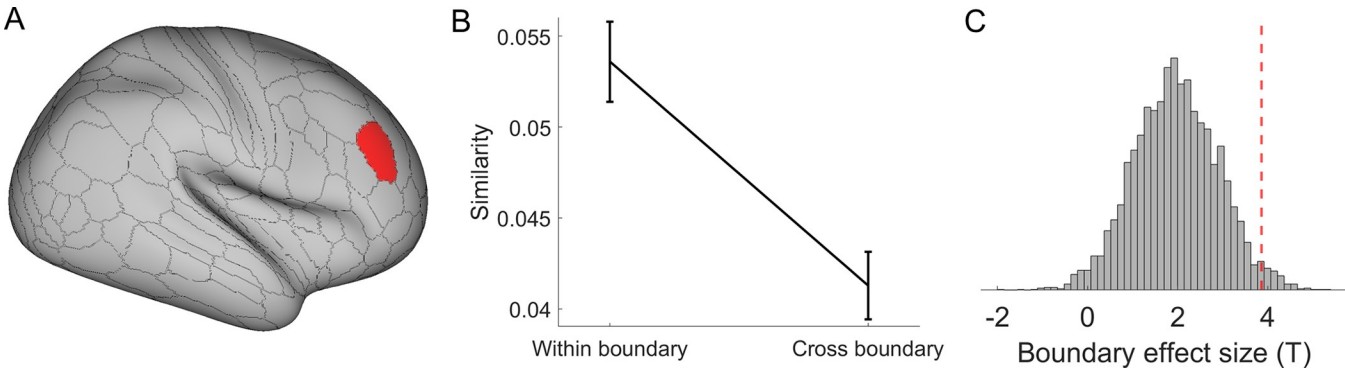

**Fig 6. Dorsolateral PFC encodes the boundary effect across the transition point.** (A) The right p9-46v shows a higher within-boundary than cross-boundary pattern similarity for trials around the transition points. (B) Pattern similarity (group mean ± SEM) as a function of within/cross-boundary in the region right p9-46v. Data used for (B) can be found in S1 Data, specifically in the sheet labeled "Fig 6B." (C) The significant boundary effect (i.e., the higher pattern similarity within-boundary than cross-boundary condition, indicated by the red dashed line), compared with a null distribution obtained by randomly shuffling the transition point 5,000 times.

values, defined as a weighted sum of their costs and benefits. Conceptually, the rule-based strategy retrieves a task before the cued position (e.g., the first task of a sequence) and recalls later tasks using sequential memory until the cued position is reached. On the other hand, the memory-based strategy directly retrieves the cued task through cue-task associations. Thus, the rule-based strategy is more effortful (and thus costly) than the memory strategy. Early in the test phase, the cue-task association is weak, whereas the participants have strong sequential memory of the task order from the training phase. Thus, the rule-based strategy provides better value than the memory strategy due to its higher benefit (despite higher cost) and is favored. As participants receive feedback with practice, the cue-task association becomes stronger. Along with its smaller effort compared to the rule-based strategy, the memory-based strategy eventually achieves a higher value, leading to the strategy transition (Fig 1A, left panel).

Based on the trial-wise model prediction of which strategy to use, trials were categorized as rule-based and memory-based trials. For each participant, the model outputs were then linked to trial-wise RT using a linear model (M0, Table 1) with the following regressors: (1) rule implementation cost (C, number of steps from recalling start to cued position) for rule trials;

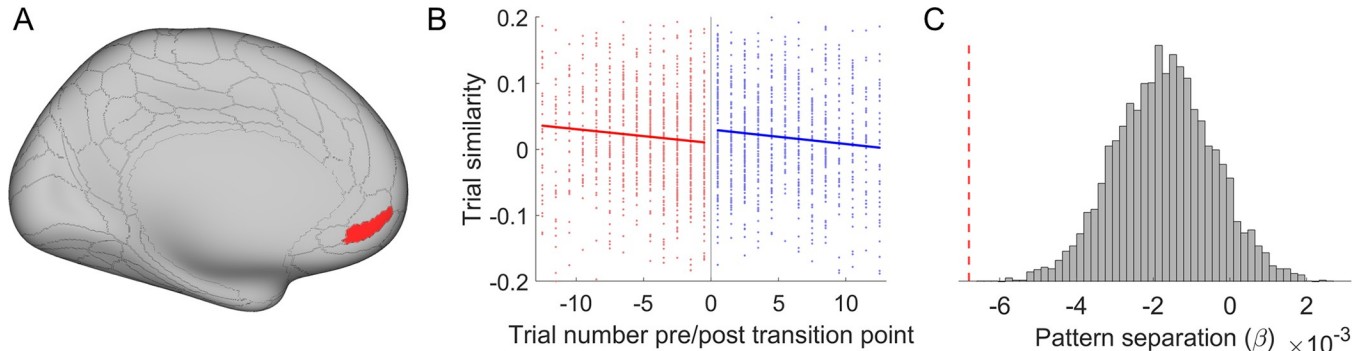

**Fig 7. Ventromedial PFC pattern-separates cue-task associations.** The left 10r area shows that pattern similarity among different cues decreases over time. (A) The anatomical location. (B) A scatter plot of trial similarity as a function of trial number relative to the transition point and the corresponding fitting lines. Red and blue colors indicate rule and memory trials, respectively. Data used for (B) can be found in S1 Data, specifically in the sheet labeled "Fig 7B." (C) The significant pattern separation effect (i.e., the negative coefficient, indicated by the red dashed line) compared with a null distribution obtained by randomly shuffling the transition point 5,000 times.

(2) the strength of the cue-task association (S) for memory trials, quantified as the reversed entropy of the task distribution conditioned on the cued position; (3) trial type (i.e., rule trials ($Type_R$) versus memory trials ($Type_M$)); (4) the interaction between C and aggregated times of rule implementation since the beginning of the experiment ($C*Times_R$) reflecting the practice effects of rule implementation (i.e., more experience would accelerate sequence recall and reduce the effect of C on RT); (5) log-transformed trial number reflecting practice effect not involving strategy switch; and (6) response (dummy coded). We compared M0 to its reduced versions (Table 1, M1-M5) using likelihood ratio tests at the group level. Specifically, M1 assumes behavioral performance does not vary across the 2 trial types; M2 assumes there is no practice effect of rule implementation; M3 incorporates both assumptions from M1 and M2; M4 assumes only a rule-based strategy has been implemented throughout the task; and M5 assumes the model uses neither rule-based nor memory-based strategies. M0 outperformed all other models ($ps < 0.001$, Fig 2C). Of note, M4 and M5 were outperformed by all other models that assume 2 strategies ($ps < 0.001$), suggesting that both strategies were used. These results suggest that all components in M0 are important to account for behavior.

We then compared M0 to 3 extended models, each examining the necessity of accounting for a specific factor. First, to test whether the behavioral performance considers the improved efficiency of rule-based strategy (i.e., increased instead of fixed values) in the cost-benefit analysis, we compared M0 to M6 (Table 1 and Note B in S1 Text). Comparison between unnested models was achieved through the Akaike information criteria (AIC). The results showed that M6 (AIC = –3,928) performed worse than M0 (AIC = –4,019), $p < 0.001$, suggesting no evidence of such consideration. Second, we explored the possibility of applying a backward sequence recall strategy with model M7, where recall could start from the last position and progress backward to the cued position, akin to backward replay. This model (AIC = –3,984, worse than that of M0, $p < 0.001$) was not supported by behavioral data (Table 1 and Note C in S1 Text). Third, we tested whether the cost-benefit analysis was probabilistic (i.e., strategy can switch back and forth) using M8, which assumes that both strategies influence trial-level performance in a weighted manner (see Materials and methods). M8 performed worse (AIC = –3,095) than M0 (Table 1, $p < 0.001$), suggesting that only the winning strategy determines behavioral performance.

Finally, to compare M0 to the decision-memory-threshold model, we developed another model (M9) without the cost-benefit analysis. This model predicts strategy switches when the memory strength reaches a threshold, regardless of the value of the rule-based strategy. However, M9 performed worse (AIC = –3,823) than the M0, $p < 0.001$ (Table 1), suggesting that the decision relies on the evaluation and comparison of alternative strategies.

As M0 was the winning model, it was used for all subsequent analyses. Individual model parameters from M0 are shown in Fig 2D. The effect of C was significantly above 0 ($\beta = 0.33$, SE = 0.09, Cohen's d = 0.64, 95% CI = [0.15, 0.51], $t(32) = 3.70$, $p < 0.001$), indicating that participants were slower when the model predicted more steps to recall. The interaction between the implementation cost and the accumulated rule-based trials was significant ($\beta = –0.09$, SE = 0.04, Cohen's d = –0.38, 95% CI = [–0.18, –0.01], $t(32) = –2.20$, $p = 0.035$), suggesting that sequence recall accelerated over time. The memory strength effect (S) was also significant ($\beta = –0.61$, SE = 0.17, Cohen's d = –0.61, 95% CI = [–0.96, –0.26], $t(32) = –3.52$, $p = 0.001$), suggesting that participants responded faster when the model predicts a stronger cue-task association on memory trials. Based on this model, the predicted proportion of participants using the rule-based strategy decreased over time (Fig 2E), indicating that the rule-based strategy was replaced by the memory strategy with practice. We defined transition points as the moments immediately before the trial in which the value of the memory strategy first surpassed the rule-based strategy (S2A Fig). Transition points were defined for each cue except for cues at

position 1 because for that position no transition was necessary. Among the remaining 192 trials for cued positions 2–5, the model predicted that a participant would apply the rule-based strategy on 94 ± 37 trials, varying from 21 to 166 trials.

## Behavioral slowing following model-predicted strategy transition

To validate the transition points predicted by the model, we tested the strategy switch cost following the transition point [19]. Using an LME model, we observed a significant increase in RT after transition points ($\beta$ = 0.40, SE = 0.15, 95% CI = [0.10, 0.69], $t(80.5)$ = 2.64, $p$ = 0.010, $\eta_p^2$ = 0.15, Fig 2F). To examine further whether this effect is reliably linked to the model-predicted transition points, we repeated this analysis 5,000 times using randomly selected transition points for each cue and each subject. The observed post-switch slowing was greater than all random ones ($p < 0.001$, Fig 2G). Notably, this effect is independent of the model fitting, which only assumes that the mean RT for memory and rule trials can be different (i.e., different intercepts), without any assumptions regarding the 2 adjacent trials surrounding the transition point. We further tested the alternative explanation that the structure of the computational model introduces biases leading to artificially reported post-transition slowing. To this end, we repeated the post-transition slowing analysis using randomly sampled model parameters while keeping the architecture of the model constant 5,000 times. The median of the resulting distribution was centered around 0, and significantly lower than the observed post-switch slowing, $p$ = 0.005 (S2B Fig), thus ruling out the possibility that the model artificially introduced a post-switch slowing effect. The post-transition slowing results suggest that the model accurately predicted *when* strategy transition occurs.

## Stronger encoding of the winning strategy than the losing strategy in fMRI data

As the behavior is not affected by the losing strategy, behavioral analyses cannot test whether the losing strategy is implemented as posited in the both-active model. On the other hand, the decision-selection model predicts that fMRI activation patterns selectively encode the rule- and memory-based strategies for rule and memory trials, respectively. We conducted fMRI analyses to assess the representational strength of the 2 strategies. For the rule-based strategy, we predicted that the similarity of fMRI activation patterns across 2 trials would reflect the number of shared recalled tasks (Fig 3A and 3B). For the memory-based strategy, we tested the cue effect such that the 2 trials' activation patterns will be more similar if they share the same cue than if they do not, as the cue and its association with the task are used in the memory strategy (Fig 3A and 3C).

Results showed widespread brain regions with significant rule effects on rule trials, but not on memory trials (Fig 4A and 4B). By contrast, significant cue effects were observed in widespread regions on memory trials but only in limited visual and motor regions on rule trials (Fig 4D and 4E). To statistically test the strength of each effect across trial types while controlling for processing shared by both strategies (e.g., visual processing), we compared each effect between rule and memory trials. A stronger rule effect on rule trials compared to memory trials was observed in the striatum, frontoparietal, temporal, and occipital regions (Fig 4C, $p$s < 0.05 corrected), whereas a stronger cue effect on memory trials compared to rule trials was found in similar regions with less spatial extent (Fig 4F, $p$s < 0.05 corrected). Importantly, no regions showed a statistically significant effect in the opposite direction in either test, suggesting that the representation is likely selective. A conjunction analysis identified a double dissociation between trial type and strategy in frontoparietal, temporal, occipital, and subcortical regions (Fig 4G, see S1 Table for a detailed list of brain regions and statistics). We replicated

these findings after regressing out the difficulty of different conditions with a covariate of the RT difference in the regression model (see S6 Fig). The double dissociation showing stronger representations of the winning strategy than the losing strategy suggests that the winning strategy is selectively implemented or prioritized.

## Neural coding of key variables in the decision-selection model

We then conducted a multivariate analysis of fMRI data to test the encoding of the key variables of the computational model [20]. A similar analysis using univariate fMRI methods is presented in Note F in S1 Text and S3 Fig. For the rule-based strategy, we tested the variable of cued position, as later cued position is associated with higher rule implementation cost (e.g., more tasks to sequentially recall, higher difficulty and higher working memory load). We found that this variable was encoded in the left IFSa, $p < 0.05$, FDR corrected (Fig 5A and 5D). In contrast, for the memory-based strategy, we tested the encoding of the memory strength of cue-task associations (Eq 3) and found that memory strength was encoded in primary visual regions (bilateral V1, V2, right V3, right V4), sensorimotor regions (left primary motor area, left lateral 8B, right area 2), and temporoparietal junction regions (left PF complex, left intraparietal 2, right PFm complex, right medial intraparietal area), among others (left rostral area 6, left auditory 5 complex, right middle TE1, and right dorsal 32), all $ps < 0.05$, FDR corrected (Fig 5B and 5D). In addition, given the importance of the posterior parietal cortex in memory retrieval [21], we also tested the memory strength effect in bilateral PFm, PGs, and PGi, and results showed that 4 of the 6 regions (bilateral PGs, right PFm, and right PGs) reached significance, $ps < 0.03$, Bonferroni corrected.

To test the presence of the cost-benefit analysis process predicted by the decision-selection model, we searched for brain regions encoding the decision variable, defined as the absolute value difference between the 2 strategies. We expected to observe this effect in the dorsomedial prefrontal area [16]. Consistently, the right 8BM and the right supplementary and cingulate eye field (SCEF) showed a significant decision variable effect, $ps < 0.05$, FDR corrected. In addition, this effect was also observed in primary visual regions (bilateral V1-V4), sensorimotor regions (bilateral areas 1, 2, 3b, 4, right 3a, bilateral dorsal area 6, bilateral 24dd, right 7AL), as well as other areas such as the right anterior area 6, the right anterior part of inferior frontal junction (IFJa), right PF complex and right A4, $ps < 0.05$, FDR corrected (Fig 5C and 5D). The evidence supporting the encoding of the decision variable also argues against the both-active model, which assumes that there is no decision-making process to select the winning strategy prior to the race.

## Brain-behavioral correlation of memory-based strategy

We next examined the explanatory power of the decision-selection model by investigating whether the model-fitting results were linked to the neural findings. The model fitting yielded a λ parameter that reflects the preference for the memory-based strategy over the rule-based strategy. A smaller λ value corresponds to a relatively higher value of the memory-based strategy (see Materials and methods). Therefore, we predicted that a smaller λ would correlate with stronger encoding of memory strength in the brain. To test this prediction, we calculated the average memory encoding strength across regions showing statistically significant representation of memory strength (Fig 5B). Our analysis revealed a negative correlation between λ (log-transformed) and encoding strength, $r = -0.52$, $p = 0.001$, one-tailed, uncorrected (Fig 3E). This negative correlation was observed in 3 out of 17 regions (bilateral V1 and left PF), $ps < 0.05$, one-tailed, FDR corrected, with the strongest correlation observed in the left PF, $r = -0.50$, $p = 0.026$, one-tailed, FDR corrected (Fig 5F). These results suggest that the extent to

which individuals prioritize the memory strategy is related to their neural sensitivity of memory encoding.

## Dorsolateral PFC encodes boundary effect across the transition point

Our behavioral and modeling results suggest an abrupt shift of strategy when the value of memory retrieval outweighs the rule implementation (Figs 2F and S2A). We anticipated that there are neural substrates underlying this abrupt shift [5,22], distinct from the overall differences observed during the constant application of the 2 strategies (Fig 4). Therefore, we next aimed to identify the brain regions supporting the strategy switch. To this end, we treated a transition point as a boundary separating the 2 time periods using different strategies. We then tested the boundary effect (i.e., higher pattern similarity between trials on the same side of the boundary than different, termed as "within boundary" and "between boundary" conditions, respectively). To focus on the effects specific to the boundary, we compared pattern similarity of the trials immediately adjacent to the transition point with other temporally close trials (i.e., between 2 and 6 trials away from the transition point). We excluded trial pairs involving different cues to prevent potential confounding effects from intrinsic differences across cues (Fig 3C). We found that the right dorsolateral PFC (dlPFC region, area p9-46v) showed a statistically significant boundary effect (Fig 6, $\beta$ = 0.0109, SE = 0.0028, 95% CI = [0.0054, 0.0165], $t$ (1358.0) = 3.86, $p$ = 0.023 corrected, $\eta_p^2$ = 0.012). No other regions survived the correction for multiple comparisons. This effect was stronger than its null distribution using shuffled transition points over 5,000 iterations ($p$ = 0.024, Fig 6C), suggesting that the boundary effect is tied to the model-estimated transition points. To further rule out the possibility that this finding captured a generic trial-type effect (i.e., independent of the trials' temporal distance from the transition points), we repeated the same analysis by replacing the temporally close trials with trials that were more than 6 trials away from the transition point and did not observe the boundary effect ($t$(30.7) = 0.84, $p$ = 0.203 uncorrected). In other words, it appears that this effect only occurred on trials near strategy transition and hence supports the relation between dlPFC representation and strategy transition.

## Ventromedial PFC pattern-separates cue-task associations

As the decision-selection model considers the improvement in memory strength as the driving factor of the strategy switch, we tested the increase in memory strength by analyzing the level of pattern-separation [23]. In other words, we predicted that pattern similarity between different cues would reduce with practice. To focus on strategy transition, trials were time-locked to their corresponding transition points. Trial positions containing too few (<30) data points were excluded from the analysis. The predicted decrease of between-cue pattern similarity was observed in the left ventromedial PFC (vmPFC, 10r area, Fig 7A and 7B, $t$(52.4) = −3.93, $p$ < 0.001, $p$ = 0.048 corrected, $\eta_p^2$ = 0.23). No other regions survived the correction for multiple comparisons. Moreover, to explore the possibility that vmPFC displays a generic decrease of between-cue pattern similarity over time, we repeated this analysis 5,000 times with randomly selected transition points. These randomized data maintain the chronological order of trials. The generic decrease hypothesis predicts that the effects in the repeated analysis will be similar to the observed effect. However, results showed that the degree of pattern separation observed in the left vmPFC (i.e., the negative slope $\beta$) falls below all values in this distribution, $p$ < 0.001 (Fig 7C). These rules out a possibility of bias due to a generic temporal effect. Together, the results suggest that the vmPFC separates neural representations of different cue-task associations during task learning, potentially strengthening these associative memories via reducing interference between associations.

## Discussion

Practice improves task performance not only by making task execution more efficient but also by changing the strategy used to perform the task. Here, we investigated when and how task strategies and related neural representations change in task learning. After learning two task sequences, participants were asked to report the task type at a cued position from one sequence. The behavioral data showed worse performance at later cued positions, suggesting that participants adopted a strategy to recall (via forward replay) the task sequence to obtain an answer. Additionally, the strength of the cued position effect decreased over time, which could be explained by a transition to a memory-based strategy that retrieves the cued task directly from memory (i.e., performance is not sensitive to cued position). To understand *how* and *when* the transition occurs, we incorporated decision-making principles—cost-benefit analysis across strategies and selective strategy implementation—into a decision-selection model. This model accurately predicted transition points, as evidenced by a significant post-transition switch cost (Fig 2F). Support for the cost-benefit analysis hypothesis also came from the observation that the decision-selection model outperformed a decision-memory-threshold model, which predicts that strategy transition occurs whenever the memory strength of cue-task associations surpasses a certain threshold. We further observed fMRI activation patterns that aligned with a stronger representation of the winning strategy (Fig 4), consistent with the decision-selection model. Importantly, we observed an encoding of the decision variable (Fig 5C) that supports the decision-selection model over the both-active model. We then delved into the decision-selection model and found that strategy transition is related to the shift in neural representation in the dlPFC (Fig 6) and associated with a gradual pattern separation in the vmPFC (Fig 7).

These findings highlight *when* the transition between rule- and memory-based strategies occurs. Our computational model reconciles the gradual increase in pattern separation in neural data (Fig 7, also see ref [24]) and the abrupt strategy switch (Fig 2F, also see ref [5]) by introducing trial-level cost-benefit analysis between the 2 strategies. In other words, as the cue-task associations strengthen, the cost-benefit tradeoff (i.e., value) for the memory strategy increases gradually and monotonically (S2A Fig). When it starts to achieve a better cost-benefit tradeoff than rule implementation, subsequent trials will use the memory strategy. The estimated transition points were validated by independent analyses of post-transition slowing (Figs 2F and S2B) and the boundary effect in the dlPFC (Fig 6). These model-based findings not only documented a strategy shift in practice-induced automaticity [13], but also provided a new decision-making account of *how* it occurs. Specifically, model comparison results (Table 1) highlight the importance of cost-benefit tradeoffs in the human brain's decision-making process when choosing between alternative strategies [16,25,26]. We further observed that the decision variable of the cost-benefit analysis was encoded in the dmPFC (Fig 5C). This is consistent with the finding that dmPFC accumulates information prior to the strategy shift [5], indicating that the cost-benefit analysis may provide the information to switch strategy in our study.

Several brain areas seem to represent both strategies (Fig 4). This may seem contradictory to the assumption that there are 2 distinct strategies. We argue that although the 2 strategies lead to the response in different ways, they indeed share some common cognitive processes, such as visual processing of the cue, memory retrieval, and response production. Thus, the same brain regions may be involved in some processes that support both strategies. Crucially, we showed a double dissociation in a subset of these regions, such as the frontoparietal regions (Fig 4G), where their activation patterns changed with the strategy. This suggests that these brain regions support the 2 strategies in different ways. The double dissociation results are

consistent with the decision-selection model. One alternative explanation to this finding is that the losing strategy was implemented but terminated prematurely as the other strategy won the race, leading to weaker representations. Thus, the double-dissociation patterns in Fig 4 are not sufficient to rule out the both-active model. However, we argue that this explanation is not against the conclusion that the winning strategy is represented more strongly than the losing strategy. More importantly, the encoding of the decision variable (Fig 5C and 5D) endorses the decision-selection model.

Of note, the recall of tasks from learned sequences suggests that sequential replay is part of the rule-based strategy [27,28]. To test this hypothesis, we conducted a multivariate classification analysis to decode tasks that could be replayed (i.e., tasks prior to the cued task). We found that these tasks could indeed be decoded from regions such as the left dlPFC, but only in rule trials, not in memory trials. Additionally, in frontoparietal and visual areas, these tasks could be better decoded in rule trials compared to memory trials (Note H in S1 Text and S7 Fig), thereby supporting the assumption of forward replay through the task sequences.

When focusing on behavioral and neural changes related to strategy transition, we first observed post-transition slowing in behavior (Fig 2F) and a numerical trend of increased fMRI activation in the prefrontal cortex following estimated transition points (S4 Fig). We further found that dlPFC activation patterns were different before and after strategy transition (Fig 6). This difference was specific to trials near the transition point instead of reflecting the difference between the 2 strategies per se. This boundary effect may signal the establishment of an event boundary [22,29] that segments experiences by strategy to reduce interference from the rule-based strategy after the strategy transition [30]. We speculate that the right dlPFC may encode 2 contexts to host cognitive control information to guide the execution of different strategies. This is in line with our prior work documenting that the right dlPFC encodes context-specific cognitive control information in a task switching paradigm [30] and organizes different types of cognitive control information [31].

Pattern separation is a fundamental mechanism for organizing the memories of multiple similar items [23]. We showed that with practice, the neural pattern similarity across different cues was gradually reduced (Fig 7), implying an increasingly pronounced pattern separation. This finding is in line with recent research illustrating robust pattern separation over time during the learning process [24]. We identified the most robust pattern separation in the left ventromedial frontal region [32], underscoring the significance of the vmPFC in episodic memory [33], pattern separation [34], prospective memory [35], and concept construction [36].

In conclusion, the present study demonstrates *how* and *when* a transition from a rule-based to a memory-based strategy occurs during practice to improve performance and reduce effort. The strategy transition occurs when the value of a memory-based strategy outweighs that of the rule-based strategy, as determined by a cost-benefit analysis. This process is characterized by a selective representation of the more efficient strategy at any given time, which, while optimizing performance, introduces an event boundary and a cost of transition between different strategy phases. These findings offer a new decision-making framework for understanding how practice is associated with neural and behavioral changes.

## Materials and methods

### Subjects

We enrolled 36 participants. Two subjects were excluded due to lower than 70% accuracy in the test phase (see below). One more subject was excluded due to excessive head motion. The final sample consisted of 33 participants (18 to 42 years old, average of 25.0 ± 6.9 years; 22 females). All participants reported no history of psychiatric or neurological disorders, with

normal or corrected-to-normal vision. The experiments were approved by the Institutional Review Board of the University of Iowa (Approval No.: 202001312). Written informed consent was obtained from all subjects prior to the experiment. The study was conducted according to the principles expressed in the Declaration of Helsinki.

### Experimental design

**Training phase.** The purpose of the training phase is to provide the participants with a set of rules (i.e., task sequences) to construct a rule-based strategy. Participants underwent training via an online web-based program to learn task transitions in sequences A (a→b→c→d→e) and B (d→c→a→e→b) (Fig 3A). They performed multiple 5-trial mini-blocks to equip an avatar with gear based on a goal image (Fig 1B). Each trial started with a sequence cue, followed by a goal image, a gear selection screen, and feedback. To facilitate the learning of task transitions, 2 test trials were administered after each mini-block, in which participants indicated the 1st/2nd/3rd task following a cued task (S1 Fig). The training lasted 3 to 6 blocks and stopped when participants achieved a block-average accuracy of >90% on test trials. Participants underwent the above training between 4 and 24 h before scanning and performed another short repetition (1 to 2 blocks) immediately before scanning. For a detailed description, see Note D in S1 Text.

**Test phase.** In the test phase, the participants performed a new task that can be initially solved using the rules learned in the training phase. With practice, the task can also be solved using the memory-based strategy. The test phase was conducted in the scanner. The stimuli were back-projected onto a screen (with a viewing angle of ~14.0˚ between the equipment pictures and the center of the screen) behind the subject and viewed via a surface mirror mounted onto the head coil. On each trial, a cue with a letter and a number indicating the sequence and position (e.g., A4) was shown at the center of the screen (Fig 1C). Six equipment images, representing the 5 equipment types and a foil (MOUSTACHE), were shown around the cue. The images were randomly selected from the 2 images of each equipment type. The locations of the 6 equipment types were randomized at the subject level but fixed within each subject. These fixed locations facilitate the implementation of the memory-based strategy. Participants were asked to select the equipment type representing the cued task with a pair of button pads, each assigned with three fingers (index, middle, and ring fingers). The screen proceeded to a fixation for 8 s if participants responded correctly or to feedback for 1 s and then a fixation for 7 s if an error was made or no response was received within 9 s following the cue. The test phase consisted of 6 blocks of 40 trials each.

**Behavioral analysis.** The analysis was centered on test phase data with RT and ER as dependent variables. For RTs, we excluded the error trials (4.4%) and trials with outlier RTs (>3 SDs above the median, 1.5%). We constructed an LME model with the block (1–6), position (1–4, which enabled the isolation of the clean effect of sequential recall by excluding position 5—known to involve a recency effect [37]), their interaction, and intercept as fixed effects. Each slope and the intercept also included a random effect at the subject level. We used the "AOCTOOL" and "multcompare" functions in Matlab to compare the slopes of position between different blocks. For all LME models conducted in this study, we estimated the degree of freedom with the Satterthwaite approach [38]. For validation purposes, we expanded the analysis to include positions 1–5 (see Note A in S1 Text).

**Computational model.** We built a computational model to simulate how the brain chooses between rule- and memory-based strategies using cost-benefit analysis. For the cost, the rule-based strategy has a cost C compared to the memory strategy:

$$C = L_t - i, \tag{1}$$

where $L_t$ is the cued position (e.g., 4 in cue A4), and $i$ is the sequence recall starting position. For example, when $i$ is 1 and $L_t$ is 4, the cost is 3 as there are 3 task transitions from position 1 to 4.

We operationalize the benefit as the probability of generating the correct response. For the rule-based strategy, as the participants have learned the transitions between steps from the training phase, the benefit depends on the strength of cue-task association at the starting position of the recall. For the memory strategy, the benefit relies on the strength of cue-task association at the cued position. Thus, for each sequence, we modeled the cue-task association using a probabilistic distribution for each position $i$ on each trial $t$ as $M_{i,t}(j)$, which represents the model's belief that the task $j$ is at position $i$. At the beginning of the experiment (i.e., $t = 0$), $M_{i,t}(j)$ was initialized in the following manner: For position 1, we assumed a perfect memory from the training phase with a probability of 1 for the correct task. For positions 2–4, a uniform distribution was used to simulate the situation of no informative cue-task associations. In position 5, the probability of the correct task received a boost of $\gamma$ to account for the recency effect; all other tasks had the same probability. After each trial, the probability distribution $M_{i,t}$ is updated as a result of learning the correct task at the cued position using a delta rule:

$$M_{i,t+1}(j) = \begin{cases} M_{i,t}(j) + \alpha*(1 - M_{i,t}(j)), & \text{if } j \text{ is the true task at position } i \\ M_{i,t}(j) - \dfrac{\alpha*(1 - M_{i,t}(j))}{4}, & \text{if } j \text{ is another task} \end{cases}, \quad (2)$$

where $\alpha$ is the learning rate ranging from 0 to 1.

The memory strength index S is quantified as the reversed entropy of $M_{i,t}$:

$$S = -H(M_{i,t}) = \sum_{j=1}^{5} p_j * \log(p_j). \quad (3)$$

Note that a higher S indicates stronger memory strength. Therefore, the quantified value (i.e., cost-benefit tradeoff) for starting recall at position $i$ is:

$$V_{i,t} = \lambda * S - C = -\lambda * H(M_{i,t}) - (L_t - i), \quad (4)$$

where $\lambda$ is the tradeoff between cost and benefit, ranging from 0.01 to 100. On trial $t$, the decision of choosing the best recall starting position $i^*$ is determined by:

$$i^* = \underset{i}{\mathrm{argmax}}(V_{i,t}). \quad (5)$$

When $i^*$ is the cued position, memory strategy is implemented as no sequence recall is needed. When $i^*$ is before the cued position, a rule-based strategy is selected.

**Model fitting.** We hypothesized that the RT for the rule and memory trials would be influenced by the rule implementation cost and memory strength, respectively. Therefore, we fit the model outputs to the RT using a linear regression, which serves as the connection of the model predictions and behavior:

$$RT \sim 1 + C*Type_R + S*Type_M + Type_R + Type_M + C*Times_R + \log(Trial) + Response, \quad (6)$$

where the $Type_R$ and $Type_M$ encode the factors of rule and memory trials, respectively, while the $C*Times_R$ encodes the interaction between rule implementation cost and the aggregated number of recalled trials since the beginning of the experiment. The Response is a categorical factor encoding the response buttons. The objective function was defined as the residual sum of squares (RSS) from Eq 6. To estimate the free parameters $\alpha$, $\lambda$, and $\gamma$, we ran the optimization 100 times (using "fmincon" in Matlab) with random starting values and chose the values that led to the minimum RSS. To determine the best structure of the regression model, we

compared this model to its reduced versions (Table 1) using the likelihood ratio test. The likelihood of each alternative model was estimated using an LME model incorporating data and predictors from all subjects. *P*-values comparing the unrestricted model (M0) to each of the restricted models (M1–M5) were obtained through the ratio of likelihood, assessed against a chi-square null distribution, with degrees of freedom equal to the difference in the number of parameters between the 2 models. Comparisons between the alternative models were achieved using either the likelihood ratio test when they are nested or the AIC otherwise. The *p*-value for the AIC comparison was estimated with $\exp(-\Delta AIC/2)$ between 2 models.

To further test whether strategy selection is probabilistic or deterministic, we refined our model (M8) by incorporating a trial-wise weighting based on the certainty of the chosen strategy. Specifically, the weight W was calculated using a softmax function:

$$W = \frac{e^V}{e^{V_R} + e^{V_M}},$$

where $V_R$ and $V_M$ represent the values of rule- and memory-based strategies, respectively, while V denotes the expected value (either $V_R$ or $V_M$, determined by the trial type) of each trial. As M8 predicts which strategy to use in a probabilistic manner, we used the probabilities as observation-level weights to predict RT using the same regression formula in Eq 6. The regression analysis predicting behavioral RTs was conducted with the "lscov" function in Matlab. Other settings are the same as M0.

The decision-memory-threshold hypothesis was tested with a new model (M9). Unlike previous models that decide between rule- and memory-based strategies, M9 specifically predicts that a transition to memory-based strategy occurs when its value exceeds a certain threshold (i.e., without comparing to the rule-based strategy). Therefore, we introduced a subject-specific free parameter $\theta$ (indicating the threshold), against which we compare the value of the memory-based strategy on each trial. This comparison determines whether to retain the rule-based strategy or switch to the memory-based strategy. In addition, we removed the parameter $\lambda$ from M9, as it does not involve direct comparisons between the 2 strategies. All other settings remain consistent with those of M0.

**Behavioral transition point analysis.** To estimate the group-level effect of post-strategy switch slowing [19], we conducted an LME model using the trial type to predict RTs for the last rule and first memory trial. An intercept was added to capture the average RT. All predictors are treated as both the fixed effect and random effect at the subject level. Therefore, the model takes the following form:

$$RT = 1 + TrialType + C + S + C*Times_R + \log(Trial) + RandomSlopes + \varepsilon, \tag{7}$$

where TrialType encodes either rule or memory trials, and RandomSlopes represents the random effect for the intercept and the slope of each fixed effect. We also estimated the null distribution for the post-strategy switch slowing by repeating the LME analysis using random permutation tests, in which transition points were shuffled 5,000 times. To test if the model inherently introduces the post-strategy switch slowing, we ran an additional permutation test by randomizing the model parameters 5,000 times. The *p*-value was determined by the percentage of permutations inconsistent with the null hypothesis (e.g., with stronger or the same post-switch slowing than the real data in this test).

**fMRI Image acquisition and preprocessing.** Functional imaging was performed on a 7T GE 950 scanner using echo-planar imaging (EPI) sensitive to BOLD contrast [in-plane resolution of $1.7 \times 1.7$ mm$^2$, $128 \times 128$ matrix, 75 slices with a thickness of 2.0 mm and no interslice skip, repetition time (TR) of 1,000 ms, multiband acceleration factor of 4, echo-time (TE) of

20.5 ms, and a flip angle of 50˚]. In addition, a sagittal T1-weighted anatomical image was acquired as a structural reference scan, with a total of 512 slices at a thickness of 0.43 mm with no gap and an in-plane resolution of $0.70 \times 0.43$ mm$^2$.

Before preprocessing, the first 8 volumes of the functional images were removed due to the unstable signals. The anatomical and functional data were preprocessed with the fMRIprep 22.0.2 [39] (RRID:SCR_016216), which is based on Nipype 1.8.5 [40] (RRID:SCR_002502). The BOLD time series were resampled to the MNI152NLin2009cAsym space without smoothing. For a more detailed description of preprocessing, see Note E in S1 Text. After preprocessing, we resampled the functional data to a spatial resolution of $2 \times 2 \times 2$ mm$^3$. The preprocessed data was smoothed with a full width at half maximum (FWHM) Gaussian kernel of 6 mm and 2 mm for univariate and multivariate analyses, respectively. All analyses were conducted in volumetric space, and surface maps are produced with Connectome Workbench (https://www.humanconnectome.org/software/connectome-workbench) for display purposes only.

**Estimation of fMRI activation.** A generalized linear model (GLM) was conducted across 6 sessions (one for each block), with each session comprising trial-wise regressors time-locked to the onset of the cue. These regressors were convolved with a canonical hemodynamic response function (HRF) in SPM 12 (http://www.fil.ion.ucl.ac.uk/spm). We further added volume-level nuisance regressors, including the 6 head motion parameters, global signal, white matter signal, cerebrospinal fluid signal, and outlier TRs (defined as standard DVARS >5 or frame-displacement >0.9 mm from previous TR). Low-frequency signal drifts were filtered using a cutoff period of 128 s. The GLM yielded 1 beta map per trial, which was then noise normalized by dividing the original beta coefficients by the square root of the covariance matrix of the error terms [41]. The noise normalization was done separately for each ROI based on the extended Human Connectome Project multimodal parcellation atlas (HCPex, version 1.1 [42]), resulting in 1 fMRI activation map for each trial. Error trials, outlier RT trials, and trials including large head movement volumes were excluded from further analysis. ROIs containing less than 50 voxels (i.e., 37 subcortical regions) were excluded from further analyses.

**Multivariate regression analyses.** We adopted a multivariate regression analysis approach from previous research [20] to quantify the representational strength of the cued position (representing rule implementation cost), memory strength, and the decision variable. The decision variable was operationalized as the absolute value difference (log-transformed) in $V_{i,t}$ (Eq 4) between the 2 best values. The analysis was performed separately for each variable. In each analysis, a variable served as the main predictor, while the other 2 variables and the behavioral RT served as nuisance regressors. The analysis was conducted for each of the 426 ROIs, which underwent a 3-fold cross-validation procedure, where the data from 6 runs and the trial-wise variable time course were divided into chronological folds (2 runs per fold). Note that for the cued position effect, we only included participants who had rule trials in all blocks. For the training folds, trial-wise fMRI activity levels from each ROI were fitted to the variable time course using ridge regression [43] to prevent overfitting. The resulting weights were applied to the fMRI data in the test fold, generating a predicted variable time course. A stronger linear correlation between predicted and actual variable time courses indicated a robust representation of the variable in neural data. Following this, Fisher's transformation was applied to the 3 correlation coefficients obtained from each fold, and their mean served as the quantification of representational strength for the ROI.

For group-level analyses, we conducted a one-sample $t$ test against 0 for each ROI. Statistical results were corrected for multiple comparisons across all ROIs using the false discovery rate (FDR) method at the level of $q < 0.05$, with each individual ROI at uncorrected $p < 0.001$, both one-tailed (right).

**RSA.** For each ROI, the representational similarity between trials from different runs was defined as their voxel-wise correlation coefficient. Trial pairs from the same run were excluded due to their inflated similarity scores [44,45]. Vectorized representational similarity data were used as the dependent variable in the LME model analysis. In addition to fixed effects of interest, we added the intercept and 2 nuisance fixed effects encoding ROI-mean activation of each of the 2 trials to rule out the influence of activation strength on similarity score. Random effects consisted of the intercept and the slope for each fixed effect at the subject level. Statistical results were corrected for multiple comparisons across all the ROIs with FDR at the level of $q < 0.05$, with each individual ROI at uncorrected $p < 0.001$, both one-tailed. For the detailed LME models, see Note G in S1 Text.

## Supporting information

**S1 Table. Brain regions and statistical results of the representation similarity analysis (corresponding to Figs 4 and S6).**
(PDF)

**S1 Text.** Note A. Behavioral Analysis with All Cued Positions. Note B. No Evidence Supporting the Discount of Rule-based Strategy Cost in the Cost-benefit Analysis. Note C. No Evidence Supporting Backward Replay. Note D. Detailed Task Design in the Training Phase. Note E. fMRI Data Preprocessing. Note F. Univariate Evidence for Rule- and Memory-based Strategies. Note G. Detailed RSA Methods. Note H. Evidence of Replay in Rule-based Task Retrieval.
(DOCX)

**S1 Fig. Design of the test task during the training phase.** This task aimed to test participants' memory of task transitions. They were asked to choose the 1st/2nd/3rd tasks (in this example, the 1st task) following a prompted task within one sequence (in this example, sequence "B"). Responses were made with "Q," "W," "O," or "P" buttons on the keyboard without time constraints.
(TIF)

**S2 Fig. Model estimation of strategy transition.** (A) Strategy transition based on the value (i.e., cost-benefit tradeoff) between the rule-based strategy (dashed lines) and the memory strategy (solid lines) in an exemplar sequence of a subject. The arrows indicate the transition points for different cued positions. The value for the rule-based strategy remains constant over time, thus the dashed lines are horizontal. On the other hand, cue-task associations strengthen over learning, thus the solid line increases monotonically. A strategy switch occurs when the 2 lines cross (i.e., when the value of the memory strategy becomes better than that of the rule-based strategy). Note that position 5 has a higher value for memory than for the rule-based strategies throughout all trials, so the participant applied the memory strategy from the beginning. (B) The permutation results (histogram) for the post-strategy switch slowing obtained by randomizing the model parameters. This analysis seeks to examine whether the post-transition slowing is specific to the transition points identified by the model. The red dashed line shows the coefficient derived from real data and is significantly higher than the null distribution, $p = 0.005$.
(TIF)

**S3 Fig. Frontoparietal fMRI activation for rule- and memory-based strategies.** (A) Regions showing higher fMRI activation with higher rule implementation cost (i.e., number of steps replayed). (B) Trial-wise fMRI activation in the left 8C region as a function of rule

implementation cost. Data used for (B) can be found in S1 Data, specifically in the sheet labeled "S3B Fig". (C, E) Show regions with higher trial-wise activation for stronger memory strength (i.e., cue-task associations) and higher trial-wise activation for lower memory strength (i.e., higher uncertainty), respectively. (D, F) Show the trial-wise fMRI activation in the left PF and left IFJp, respectively, plotted as a function of memory strength quantified as reversed entropy.
(TIF)

**S4 Fig. Univariate strategy transition cost.** Displayed are regions showing stronger fMRI activation on the first memory trial compared to the last rule trial, thresholded at uncorrected $p < 0.05$. The involvement of frontal regions is consistent with the switching cost hypothesis that occurs when transferring from one task to another [46,47].
(TIF)

**S5 Fig. LME structure demonstration for RSA Analysis 1 (detailed in Note G in S1 Text).** Sample data from 2 participants is plotted to illustrate how the values within the representational similarity matrices are used in the LME. In the left panel, the white cells, representing the similarities/differences in a model variable between 2 trials, are converted into a vector in the LME, while the gray-colored cells are removed due to either within-run correlation (diagonal) or duplication (upright off-diagonal). In the right panel, the first column represents z-transformed representational similarity values. Columns 2 through 13 correspond to regressors for different experimental conditions: R1, R2, M1, M2, $RSM_{Rule\_R}$, $RSM_{Rule\_M}$, $RSM_{Cue\_R1}$, $RSM_{Cue\_R2}$, $RSM_{Cue\_M1}$, $RSM_{Cue\_M2}$, $RSM_{Univoxel\_trial1}$ and $RSM_{Univoxel\_trial2}$. The participant identifier is represented in the final column.
(TIF)

**S6 Fig. Double dissociation results with models controlling for difficulty.** A significant representation of the rule effect is observed on rule (R) trials (panel A) but not on memory (M) trials (panel B). (C) A stronger rule effect on rule than memory trials (R–M) was found in frontoparietal, temporal, occipital, and subcortical regions. In contrast, the cue effect is exhibited in more regions on memory trials (panel E) than on rule trials (panel D), and regions showing a stronger memory effect on memory than rule trials (M–R) include frontoparietal, temporal, occipital, and subcortical regions (panel F). (G) Regions showing double dissociation (i.e., the conjunction of C and F) with both stronger rule effect representation on rule trials and stronger cue effect representation on memory trials. The middle panel illustrates the double dissociation with an example region (left inferior prefrontal sulcus; IFSa). Error bars are standard errors of mean. Data used for (G) can be found in S1 Data, specifically in the sheet labeled "S6G Fig". All subcortical regions are depicted in an axial slice at z = 0. The right panel illustrates the double dissociation with an example region (left IFSa).
(TIF)

**S7 Fig. Decoding evidence supporting task replay.** Both panels show regions identified with stronger decoding of replayed tasks on rule than memory trials. (A) FDR-corrected results using regions identified in Fig 4C as search regions. (B) FDR-corrected results using whole-brain ROIs.
(TIF)

**S1 Data.**
(XLSX)

## Acknowledgments

We thank Tobias Egner, Tammy Tran, Kai Hwang, Jan Wessel, Eliot Hazeltine, Atsushi Kikumoto, the members of the Cognitive Control Collaborative, and the members of the Human Perception and Performance Group at the University of Iowa for helpful discussions. We thank Bettina Bustos for helping with editing the manuscript.

## Author Contributions

**Conceptualization:** Guochun Yang, Jiefeng Jiang.

**Data curation:** Guochun Yang.

**Formal analysis:** Guochun Yang, Jiefeng Jiang.

**Funding acquisition:** Jiefeng Jiang.

**Investigation:** Guochun Yang, Jiefeng Jiang.

**Methodology:** Guochun Yang, Jiefeng Jiang.

**Supervision:** Jiefeng Jiang.

**Writing – original draft:** Guochun Yang.

**Writing – review & editing:** Guochun Yang, Jiefeng Jiang.

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
