## [Editor Report · Decision Letter 0]

23 Aug 2024

Dear Dr Yang, 

Thank you for submitting your manuscript entitled "Cost-benefit Tradeoff Mediates the Rule- to Memory-based Processing Transition during Practice" for consideration as a Research Article by PLOS Biology.

Your manuscript has now been evaluated by the PLOS Biology editorial staff and I am writing to let you know that we would like to send your submission out for external peer review.

Please note that we unfortunately have not been able to receive advice from one of our academic editors on your study (hence the delay in getting back to you, for which I would like to apologize) and have, therefore, not yet made a firm decision on whether the conceptual advance is sufficient for PLOS Biology. We will discuss this after review with one of our editorial board members and will be looking for strong reviewer support.

Once your full submission is complete, your paper will undergo a series of checks in preparation for peer review. After your manuscript has passed the checks it will be sent out for review. To provide the metadata for your submission, please Login to Editorial Manager (https://www.editorialmanager.com/pbiology) within two working days, i.e. by Aug 25 2024 11:59PM.

Kind regards,

Christian

Christian Schnell, PhD

Senior Editor

PLOS Biology

cschnell@plos.org

---

## [Decision Letter · Decision Letter 1]

4 Oct 2024

Dear Dr Yang,

Thank you for your patience while we considered your revised manuscript "Cost-benefit Tradeoff Mediates the Rule- to Memory-based Processing Transition during Practice" for consideration as a Research Article at PLOS Biology. Your revised study has now been evaluated by the PLOS Biology editors, the Academic Editor and one new reviewers, since not all reviewers from Nature Communications agreed to share their identity with us. 

In light of the reviews, which you will find at the end of this email, we are pleased to offer you the opportunity to address the comments from the Reviewer 1 in a revision that we anticipate should not take you very long. Specifically, we think that you need to reframe your study because your data/model supports an alternative hypothesis. While probably no new analyses are needed (unless you wanted to provide evidence against the reviewer's interpretation), a thorough textual rewrite is needed, including further proofreading, as the reviewer mentioned.

We will then assess your revised manuscript and your response to the reviewers' comments with our Academic Editor aiming to avoid further rounds of peer-review, although might need to consult with the reviewers, depending on the nature of the revisions.

**IMPORTANT - SUBMITTING YOUR REVISION**

*Resubmission Checklist*

*Published Peer Review*

*PLOS Data Policy*

*Blot and Gel Data Policy*

Sincerely,

Christian

Christian Schnell, PhD

Senior Editor

PLOS Biology

cschnell@plos.org

REVIEWS:

Reviewer #1: In this paper, Yang and Jiang use model comparison and neuroimaging to test a model of how people shift from using rule-based decision making to using memory-based decision making (stimulus-response associations). They find that performance on a task that can be solved using these strategies improves over time. Moreover, model comparison suggests that this happens because people are starting to use the memory-based strategy. Specifically, the model argues that as stimulus-response associations are practiced strategy, their accuracy outweighs the cost of rule-based decision making. The authors contrast these predictions against several other models, and they find that their cost-benefit model provides a better fit to the data. They use neuroimaging and behavioral analyses to argue against the idea that the shift in strategies is driven by a "horse race" model (like Logan's instance theory of automatization). Here, they show that on the first trial where the model predicts a shift towards memory participants seem to incur a switch cost. For example, they show that dlPFC encodes this transition point, and that a broad group of neural regions that encodes the decision variable (the difference between the value for the two strategies).

This is an interesting paper on an old but important question. The computational modeling is sophisticated, and so are the neuroimaging analyses. When I was asked to review this paper, the editor told me my expertise aligned with the (most critical) reviewer #3. I have read their comments, and I agree with the gist behind most of their comments. I also believe that the authors have done a thorough job in responding to them. However, I still have a few remaining concerns. Some of these are more about the structure of the paper, but I also have some theoretical concerns.

My main theoretical concern is that I feel that the horse race model as described by the authors is not a fair representation of the previous reviewer's comments. The authors interpret this model to mean that participants always perform both strategies on each trial. Therefore, when they find that there is no or reduced evidence for "rule" trials later on in the session, they claim that this goes against the prediction of the horse race model. However, I believe that this a fair characterization of a horse race model. First, Logan originally proposed this as a model of automaticity, so the assumption that both algorithms are run in full on each trial seems to be unwarranted. In Logan's (1988) own words (the emphasis is mine): 

"When the stimulus is encountered again in the context of the same goal, some proportion of the processing episodes it participated in are retrieved. The subject can then choose to respond on the basis of the retrieved information, if it is coherent and consistent with the goals of the current task, [or to run off the relevant algorithm] and compute an interpretation and a response."

So, even Logan didn't belief that both strategies were active on each trial. Moreover, even if one did believe that both strategies are started, then an easy assumption is that the "losing" one is terminated when the other one wins. Thus, I do not think that the race horse model in this paper (depicted in Figure 1A) is a fair version of the model. I believe that many of the results in this paper that are presented as evidence against the race horse model are consistent with this version of the model. It's not infeasible that on the first trial where the memory strategy wins, there is a switch cost. Moreover, the neuroimaging results that show no evidence for a rule effect on memory trials are perfectly understandable when the memory system wins the race so fast that the rule-based system has no time to seriously get going.

I agree with the previous reviewer #3 that this paper is light on evidence for a cost-benefit decision. The authors argue that each step through the sequence carries some kind of cost. This is not incompatible with the horse race model, where "deeper" sequences just take longer to complete and therefore to win. Based on the reasoning described above, I'm not sure the authors have found compelling evidence for this part of their model.

Finally, it struck me as odd that RT curves in Figure 2A do not actually become flat. As far as I understand, when the memory strategy wins, the order of the cued position should become irrelevant (each cue becomes associated with a response that can be immediately executed). This is particularly true given that the authors report evidence that a deterministic (not stochastic) model provides a superior fit. This suggests that at some point the participants switch to become fully memory based, and therefore these curves should become fully flat by the last block. Why is that not the case?

My final major comment is that the paper introduces the task and the modeling philosophy in a way that is too sparse. I'm not intimately familiar with this task and this style of computational modeling, and I needed to pour over the methods and the captions of the figures before I understood the task and modeling rationale. I understand that previous outlet may have been more restrictive in their allotted word count, but I urge the authors to move a lot of the task and modeling details back to the main text.

Finally, and I think this is something that the previous authors also commented on, there are still a lot of grammatical errors in the manuscript, as well as other displays of unconventional English. For example, the first sentence is presented in quotes, but it is not attributed to anyone. There are also several sentences where noun is plural and verb singular, or vice versa ("Strategy switch can occur", "Here, we investigated how task strategy and related neural representations change in task learning"). The authors indicate they have used Grammarly to weed out errors, but I'm afraid that the mere use of this application may not have been sufficient.

---

## [Decision Letter · Decision Letter 2]

10 Dec 2024

Dear Guochun,

Thank you for your patience while we considered your revised manuscript "Cost-benefit Tradeoff Mediates the Rule- to Memory-based Processing Transition during Practice" for publication as a Research Article at PLOS Biology. This revised version of your manuscript has been evaluated by the PLOS Biology editors, the Academic Editor and the original reviewers.

Based on the reviews and on our Academic Editor's assessment of your revision, we are likely to accept this manuscript for publication, provided you satisfactorily address the remaining points raised by the reviewers. Please also make sure to address the following data and other policy-related requests:

* We would like to suggest a different title to improve accessibility for our broad audience: "Cost-benefit tradeoff mediates the transition from rule-based to memory-based processing during practice"

* Please add the links to the funding agencies in the Financial Disclosure statement in the manuscript details.

* Please include information in the Methods section whether the study has been conducted according to the principles expressed in the Declaration of Helsinki. All research involving human participants must have been conducted according to the principles expressed in the Declaration of Helsinki.

* DATA POLICY:

Regardless of the method selected, please ensure that you provide the individual numerical values that underlie the summary data displayed in the following figure panels as they are essential for readers to assess your analysis and to reproduce it: 1A, 4G, 5D, 6B, 7B, S3B and S6G. 

* CODE POLICY

* Please include the references in the supplementary reference list in the main reference list.

We expect to receive your revised manuscript within two weeks. 

*Published Peer Review History*

*Press*

Sincerely,

Christian

Christian Schnell, PhD

Senior Editor

cschnell@plos.org

PLOS Biology

Reviewer remarks:

Reviewer #1: I commend the authors for taking my considerations seriously. This version of the paper reads great. I still caught a few language errors, but I am happy to recommend acceptance if these are taken care of:

224-226 On each trial, the model performs a cost-benefit analysis to determine which strategy to use based on their values, defined as a weighted sum of _cost and benefit_.

Maybe *their costs and benefits?*

235. As participants receives feedback 

subject verb disagreement 

254. M4 assumes only rule-based strategy 

Missing a?

255-256. M5 assumes the model uses neither rule-based nor memory-based strategy. 

Strategies? Or are there two "a"s missing here?

564-566. One alternative explanation to this finding is that the losing strategy was implemented but terminated prematurely as the other strategy winning the race, leading to weaker representations.

"as the other strategy winning the race"

---

## [Editor Report · Decision Letter 3]

16 Dec 2024

Dear Guochun,

Thank you for the submission of your revised Research Article "Cost-benefit Tradeoff Mediates the Transition from Rule-based to Memory-based Processing during Practice" for publication in PLOS Biology. On behalf of my colleagues and the Academic Editor, Raphael Kaplan, I am pleased to say that we can in principle accept your manuscript for publication, provided you address any remaining formatting and reporting issues. These will be detailed in an email you should receive within 2-3 business days from our colleagues in the journal operations team; no action is required from you until then. Please note that we will not be able to formally accept your manuscript and schedule it for publication until you have completed any requested changes.

While you attend to the requests to come, please also add a statement to the corresponding figure legends where the source data can be found For example: 'Source data can be found in S1 data.'

PRESS

Sincerely, 

Christian

Christian Schnell, PhD

Senior Editor

PLOS Biology

cschnell@plos.org